# Single cell analysis of kynurenine and System L amino acid transport in T cells

Linda V. Sinclair [1], Damien Neyens[1,4], George Ramsay [2,3], Peter M. Taylor [1] & Doreen A. Cantrell [1]

The tryptophan metabolite kynurenine has critical immunomodulatory properties and can function as an aryl hydrocarbon receptor (AHR) ligand. Here we show that the ability of T cells to transport kynurenine is restricted to cells activated by the T-cell antigen receptor or proinflammatory cytokines. Kynurenine is transported across the T-cell membrane by the System L transporter SLC7A5. Accordingly, the ability of kynurenine to activate the AHR is restricted to T cells that express SLC7A5. We use the fluorescence spectral properties of kynurenine to develop a flow cytometry-based assay for rapid, sensitive and quantitative measurement of the kynurenine transport capacity in a single cell. Our findings provide a method to assess the susceptibility of T cells to kynurenine, and a sensitive single cell assay to monitor System L amino acid transport.

[1] Division of Cell Signalling and Immunology, School of Life Sciences, University of Dundee, Dundee DD1 5EH, UK. [2] Department of General Surgery, Aberdeen Royal Infirmary, Aberdeen AB25 2ZN, UK. [3] Rowett Institute, University of Aberdeen, Aberdeen AB25 2ZD, UK. [4] Present address: Ludwig Institute for Cancer Research, de Duve Institute, Université Catholique de Louvain, 1200 Brussels, Belgium. Correspondence and requests for materials should be addressed to L.V.S. (email: l.v.sinclair@dundee.ac.uk) or to D.A.C. (email: d.a.cantrell@dundee.ac.uk)

K ynurenine, the product of indoleamine 2,3, dioxygenase (IDO)-mediated catabolism of tryptophan, is a potent immunomodulatory molecule that can control T-cell immune responses[1–4]. IDO expression is strongly induced in antigen-presenting cells, especially dendritic cells, in response to inflammatory signals, including LPS, type I interferons (IFNα/β), type II interferons (IFNγ) and interleukin 1 (IL-1), as well as in response to CTLA-4-mediated signalling[5–7]. The expression of IDO is also increased in cancer cells[8, 9]. Multiple studies using genetic or pharmacological manipulation of IDO signalling have highlighted an immunomodulatory role of IDO expression to restrain inflammation and promote tolerance[5, 6]. Cells that express high levels of IDO deplete the microenvironment of tryptophan and replace it with its metabolite kynurenine. Although the depletion of tryptophan from the microenvironment is immunosuppressive[6, 10–12], kynurenine itself also has immune modulatory properties. For example, it can function as a ligand for the aryl hydrocarbon transcription (AHR) factor complex to promote effector CD4+ T-cell differentiation. In particular, AHR signalling has been shown to influence the differentiation of activated CD4+ T cells to Foxp3 expressing, immunosuppressive regulatory T cells[13, 14].

The AHR can also be triggered by dioxins such as 2,3,7,8-tetrachlorodibenzo-p-dioxin (TCDD) and by the tryptophan photo-metabolite 6-formylindolo[3,2-b]carbazole (FICZ). The concentrations of these ligands needed to activate the AHR are in the pM/nM range compared with μM levels of kynurenine[15, 16]. Why are such high concentrations of kynurenine needed to activate the AHR? In this context, Seok et al.[16] have shown that kynurenine acts more like a pro-ligand, requiring further 'activation' for AHR regulation. Hence, trace element condensation products of kynurenine can be potent AHR agonists (in the picomolar range)[16]. One other possible explanation for the relatively high kynurenine concentrations required for AHR triggering, compared with other ligands, could be differential membrane transport requirements for kynurenine versus other ligands, that is, active transport versus passive diffusion across the plasma membrane. Indeed, kynurenine has been shown to be transported by large neutral amino acid transporters (LAT) in rat astrocytes[17].

The question of how T cells transport kynurenine across their membrane to activate the AHR has not been addressed. Hence, the focus of the present work is to explore kynurenine transport characteristics of T cells using both conventional radiolabelled kynurenine uptake assays and a novel flow cytometry-based assay that utilises the spectral fluorescent properties of kynurenine. This flow cytometry assay enables rapid, sensitive and quantitative measurement of kynurenine transport by single cells. These studies show that kynurenine transport is restricted to in vitro and in vivo immune-activated T cells; the kynurenine transport capacity of naive quiescent T cells is low. Important triggers to induce kynurenine uptake are engagement of the T-cell antigen receptor (TCR) or exposure of TCR-activated T cells to proinflammatory cytokines such as interleukin-2 (IL-2). We establish that kynurenine is transported across the cell membrane of activated T cells by System L transporters and identify SLC7A5 as the critical transporter. One role for kynurenine is to be a ligand for the AHR in CD4+ T cells[15, 18, 19]. We show that this capacity of kynurenine to function as an AHR ligand in immune-activated CD4+ T cells is highly regulated and dependent upon System L transporter availability.

Kyn) into T cells. The data show that naive T cells do not effectively take up 3H-Kyn (Fig. 1a). However, TCR triggering of T cells induced a substantial increase in 3H-Kyn transport, which is efficiently out-competed in the presence of excess, non-radiolabelled kynurenine. TCR-primed CD8+ and CD4+ T cells cultured in interleukin-12 (IL-12) and IL-2 clonally expand and differentiate to cytotoxic T cells (CTL) and T helper 1 cells (TH1s), respectively. Moreover, CTLs and TH1s maintained in IL-2 also show high levels of kynurenine uptake (Fig. 1a). Activated T cells express many amino acid transporters, consequently we used a pharmacological reductionist approach to define which of these are responsible for kynurenine transport. The data in Fig. 1b show that 3H-Kyn uptake in TCR-activated CD4+ T cells, TH1s and CTLs is sodium independent, in contrast to 3H-Gln uptake which is dependent on sodium. Sodium-independent amino acid transport is a hallmark of System L transporters whose substrates include leucine, phenylalanine, methionine and tryptophan. Transport of the known System L substrate tryptophan, as well as the tryptophan metabolite kynurenine, is inhibited by the System L blocker BCH in TH1 cells and CTLs (Fig. 1c, d). The selectivity of BCH action is evidenced by its failure to block radiolabelled glutamine uptake in activated T cells (Fig. 1e).

These experiments show that populations of in vitro activated but not naive T cells have high kynurenine transport capacity. A key question is whether immune activation of T cells in vivo causes T cells to increase kynurenine transport capacity. However, addressing this question is difficult because immune-activated T cells in vivo are found at low frequency in secondary lymphoid tissue and thus are not readily amenable to analysis with conventional radiolabelled amino acid tracer assays which monitor changes at a total cell population level. The capacity to identify changes in subpopulations in complex mixtures of cells is best addressed by developing single cell assays for kynurenine uptake. In this context, a physical property of kynurenine is that it is fluorescent with an excitation wavelength of 380 nm and an emission spectrum of 480 nm; standard wavelengths for fluorophores used in flow cytometry[20, 21]. Accordingly, we explored the possibility of monitoring the capacity of single cells to transport kynurenine using flow cytometry.

In initial experiments, we used effector CD8+ CTLs to test the potential of monitoring kynurenine uptake by flow cytometry. Figure 2a shows the fluorescence of CTLs measured using a BP filter 450/50 with 405 nm laser excitation as they are exposed to kynurenine. Data were collected for 120 s to determine the baseline fluorescence of CTLs prior to addition of 200 μM kynurenine, as indicated by the red arrow (left panel). The middle panel shows the same data plotted as a trace graph of the geometric mean of the cell population against time. The data show that upon kynurenine addition, the 450 nm fluorescence emission of CTLs increases substantially. The right panel compares the 450 nm fluorescence of CTLs incubated in the presence or absence of kynurenine for 4 mins. These data show increased fluorescence over time, indicating uptake of kynurenine by the CTLs. Importantly, the estimated $K_m$ for the initial rate of kynurenine transport in CTLs using the radiolabelled kynurenine transport assay versus the flow cytometry assay were comparable (213 μM ± 70 μM versus 267 μM ± 20 μM; Fig. 2b). Moreover, the increase in fluorescence of the kynurenine exposed CTLs was prevented when the cells were treated with the System L inhibitor BCH (Fig. 2c). We also compared the ability of BCH to block 3H-Kyn uptake with the ability of BCH to block Kyn-mediated increased fluorescence (MFI) in CTL, TH1 and 24 h TCR-stimulated T cells, over 4-m uptake periods. The data in Table 1 show that BCH blocks 80–90% of radiolabelled kynurenine uptake, and 80% of kynurenine transport as measured by flow cytometry. System y + L transporters share many substrates with System L

## Results

### Antigen receptor and cytokines regulate kynurenine transport.
To characterise kynurenine membrane transport by T lymphocytes, we monitored the uptake of radiolabelled kynurenine (3H-

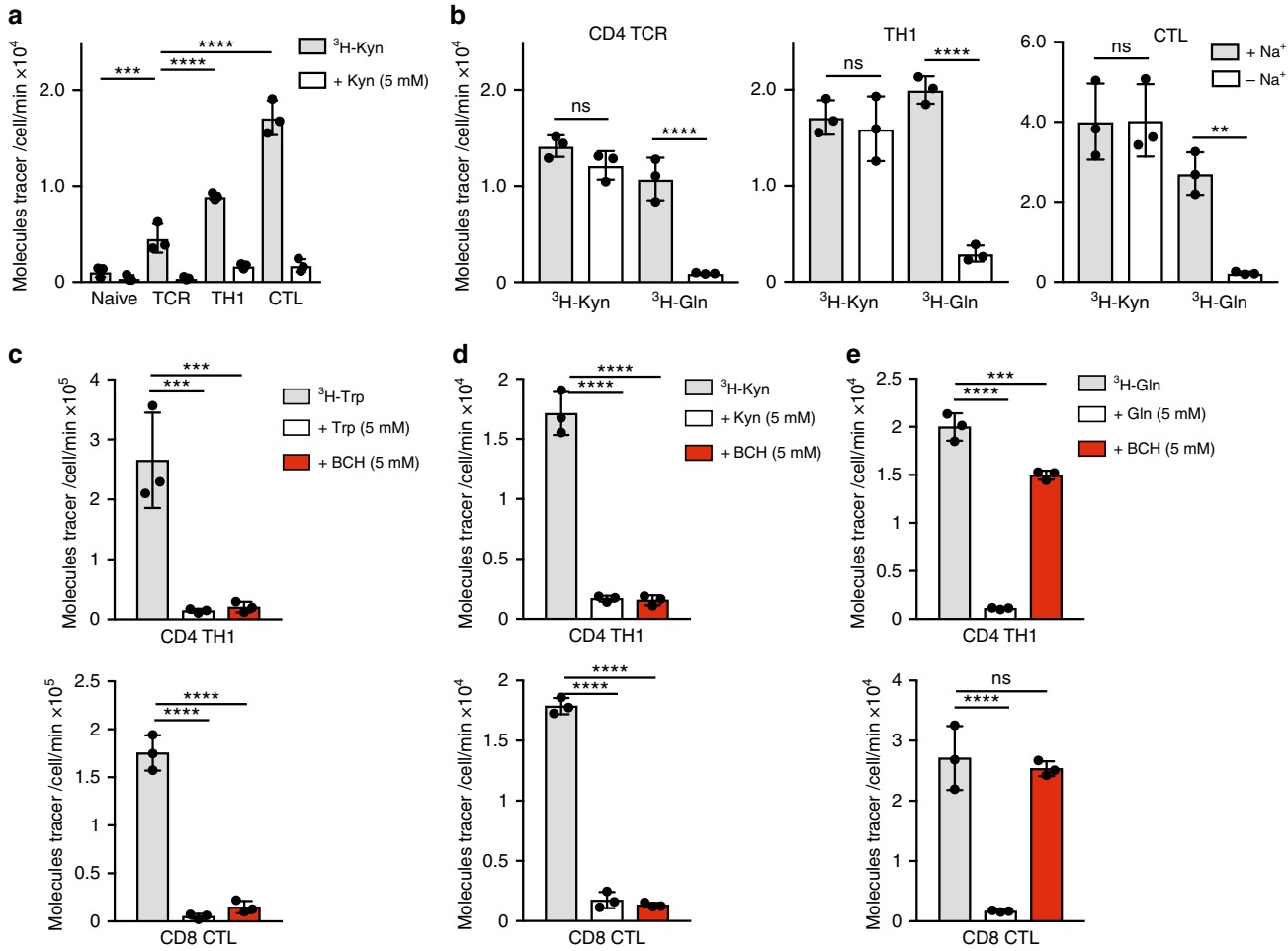

**Fig. 1** T cells regulate kynurenine transport through a System L transporter. **a** Uptake of $^3$H-kynurenine (0.1 μCi/ml, 4 min, 37 °C) in purified CD4$^+$ T cells ± TCR activation using CD3/CD28 antibodies for 18 h; in vitro expanded TH1 cells (5 days) and in vitro expanded CTLs (5 days). Cold competition with non-radiolabelled kynurenine is shown as a control for uptake specificity. **b** Uptake of $^3$H-kynurenine (0.1 μCi/ml, 4 min, 37 °C) or $^3$H-glutamine (0.5 μCi/ml, 4 min, 37 °C) performed in Na$^+$ containing or Na$^+$-free uptake buffer. The graphs show radiolabelled tracer uptake in purified CD4$^+$ T cells 48 h post TCR activation using CD3/CD28 antibodies (left panel); IL-2 maintained TH1 cells (centre panel) and IL-2 maintained CTLs (right panel). **c–e** Radiolabelled tracer uptake in IL-2 maintained TH1 cells (top panels) and IL-2 maintained CTLs (bottom panels) in the presence of non-radiolabelled substrate (5 mM; cold competition) or the System L inhibitor, BCH (5 mM). The graphs show $^3$H-tryptophan (**c**), $^3$H-kynurenine (**d**) and $^3$H-glutamine (**e**). Error bars are ±s. d. Individual points represent biological replicates. **a–e** Three biological replicates. $P$ values * = < 0.01; ** = < 0.005; *** = < 0.001; **** = < 0.0001; ns = not significant (ordinary one-way ANOVA)

transporters[22]. System L transport can be competitively blocked using excess leucine, whereas System y + L is preferentially blocked by excess lysine[23]. Accordingly, we compared the kynurenine transport after 4 min in the presence or absence of excess leucine, lysine or BCH. The data show that lysine does not affect kynurenine transport in TCR-stimulated CD8$^+$ T cells or in effector CTLs, whereas excess leucine blocks kynurenine transport to comparable levels as BCH (Fig. 2d). Together, these data indicate that kynurenine transport in activated T cells is mediated by a sodium independent, BCH-sensitive System L transporter. In parallel experiments to validate the use of kynurenine as a model substrate for System L amino acid transport, it was noted that the kynurenine-mediated increase in MFI is temperature sensitive; CTLs incubated with kynurenine at 37 °C for 4 min increase 450 nm fluorescence; this is inhibited at 4 °C (Fig. 2e). This is consistent with kynurenine being actively transported across the cell membrane. One further important observation with regard to development of a single cell assay is that the kynurenine fluorescence survives fixation with no loss of mean fluorescence

intensity (MFI) seen in fixed samples compared to live samples (Fig. 2f), thus allowing for kynurenine uptake on multiple samples to be simultaneously performed, fixed and subsequently analysed together. Moreover, kynurenine uptake in different populations within the same sample can be compared using standard flow cytometry gating strategies (Supplementary Fig. 1).

**Interrogating System L transport in cells activated in vivo**. The single cell flow cytometry assay for kynurenine uptake should permit the interrogation of the kynurenine transport capacity of rare populations of cells including antigen specific lymphocytes responding to immune stimuli in vivo. Accordingly, using this newly optimised transport assay, we investigated the ex vivo kynurenine transport capacity of in vivo antigen activated TCR transgenic OT2 CD4$^+$ T cells (Supplementary Fig. 1) and that of effector CD8$^+$ T cells generated during an immune response against a recombinant strain of *Listeria monocytogenes* (rLM). The data in Fig. 3a show that the proportion of CD8$^+$ T cells present in the spleen of rLM-infected mice is increased at D7

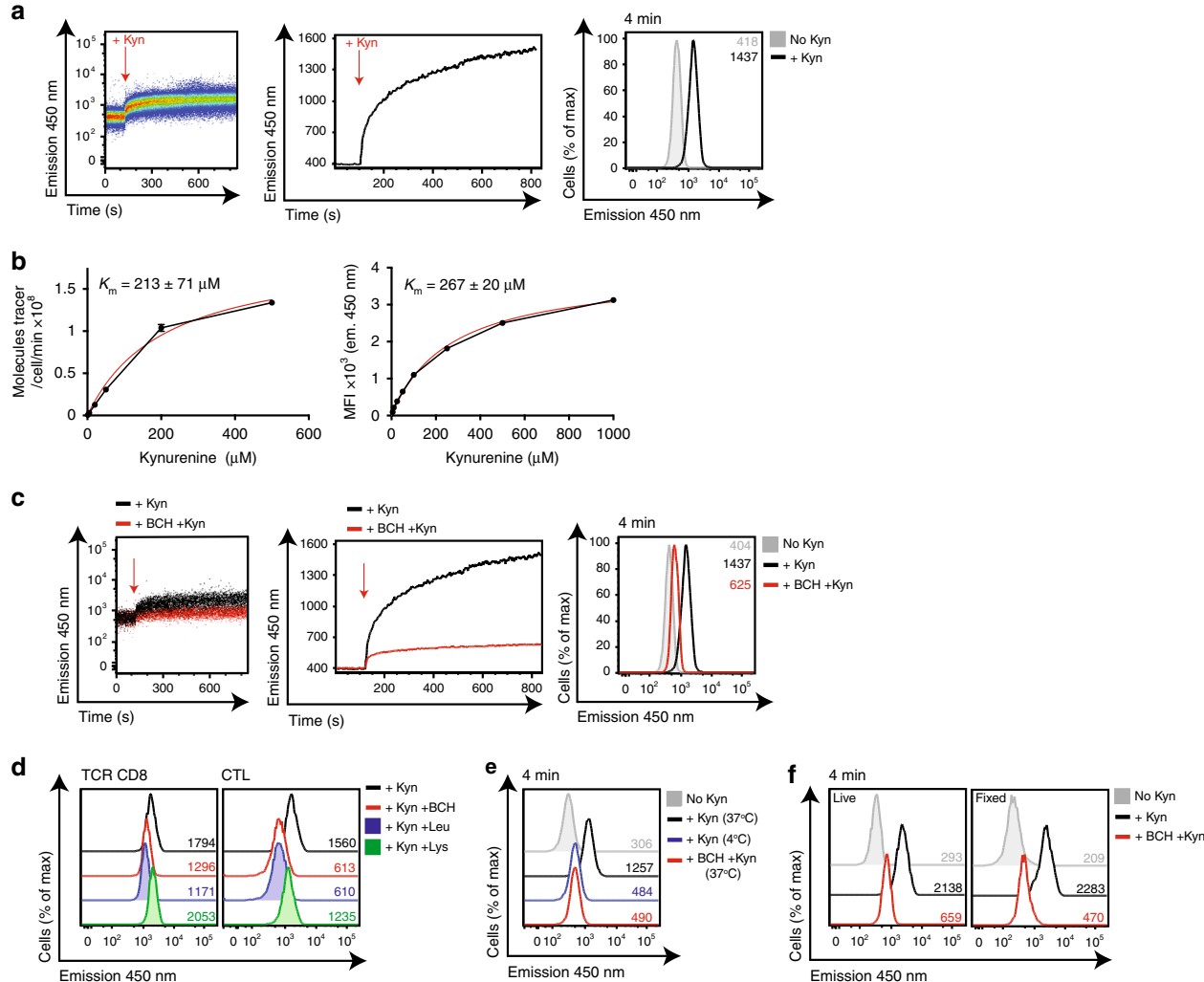

**Fig. 2** Flow cytometric monitoring of System L-dependent kynurenine uptake. **a** Flow cytometric evaluation of kynurenine uptake in IL-2 maintained CTLs. Data acquired using 405 nm excitation (violet laser) and band pass filter 450 ± 50 on BD LSRII (Fortessa). The left panels show a dot plot of fluorescence (emission 450 nm) of CTLs against time. Data acquired pre (120 s) and post (+700 s) addition of 200 μM kynurenine (indicated by red arrow). The centre panel shows the same data plotted as a geometric mean trace (FlowJo software). The right panel shows overlaid histograms of the data from CTLs in the presence or absence of kynurenine (200 μM) for 4 mins (37 °C). **b** The graphs show the number of [3]H-kynurenine molecules transported per cell per minute (left), or mean fluorescence intensity (MFI, right) of IL-2 maintained CTLs when incubated with increasing kynurenine concentrations. $K_m$ values, the concentration of kynurenine that gives half maximal uptake, were calculated using standard Michaelis–Menton equation (GraphPad Prism software). **c** 450 nm fluorescence in IL-2 maintained CTLs treated with kynurenine, as in **a**, in the presence or absence of the System L inhibitor, BCH (10 mM). **d** 450 nm fluorescence emission of 36 h CD3/CD28 stimulated CD8[+] T cells (left) or IL-2 maintained CTLs (right) treated with kynurenine in the presence or absence of BCH (10 mM), Leu (5 mM) or Lys (5 mM) at (37 °C) for 4 min. **e** 450 nm fluorescence emission of IL-2 maintained CTLs treated with kynurenine in the presence or absence of BCH (10 mM), at (37 °C); or treated with kynurenine at (4 °C) for 4 min. **f** The histograms show 450 nm fluorescence in CTLs treated with kynurenine for 4 mins ± BCH (10 mM). Cells were analysed immediately (live; top panel) or after fixation (fixed; bottom panel) using 1% paraformaldehyde. MFIs are indicated in the histograms. Data shown are representative of 6 (CTL) or 4 (TCR) experiments

**Table 1 Percentage of kynurenine uptake blocked by BCH**

|  | [3]H-Kyn | Kyn MFI |
|---|---|---|
| TCR | 79.1 ± 11.3 | 77.1 ± 14.7 |
| TH1 | 90.2 ± 2.1 | 81.0 ± 9.0 |
| CTL | 92.5 ± 0.9 | 78.3 ± 4.5 |

The table shows the percentage of kynurenine uptake blocked by the System L inhibitor, BCH as measured by [3]H-Kyn uptake or 450 nm fluorescence emission. Data from CTLs, TH1 and 24 h CD3/CD28 stimulated T cells (TCR) are shown. Data (±s.d.) are pooled from from six (CTL) or four (TH1 and TCR) experiments

post-infection. This correlates with the emergence of effector CD8[+] T cells as determined by increased CD44 surface expression and the production of the effector cytokine interferon gamma (IFNγ) (Fig. 3b, c). The data show that kynurenine transport was readily detectable ex vivo in D7 rLM-infected CD8[+] T cells, and that this transport was blocked by BCH and leucine competition but not by lysine competition (Fig. 3d). Figure 3e shows the transport of kynurenine in CD8[+] T cells expressed as a ratio of kynurenine uptake (MFI) to kynurenine uptake in the presence of BCH (MFI), allowing for the direct comparison of System L-mediated uptake across the experiment. The data show that effector CD8[+]

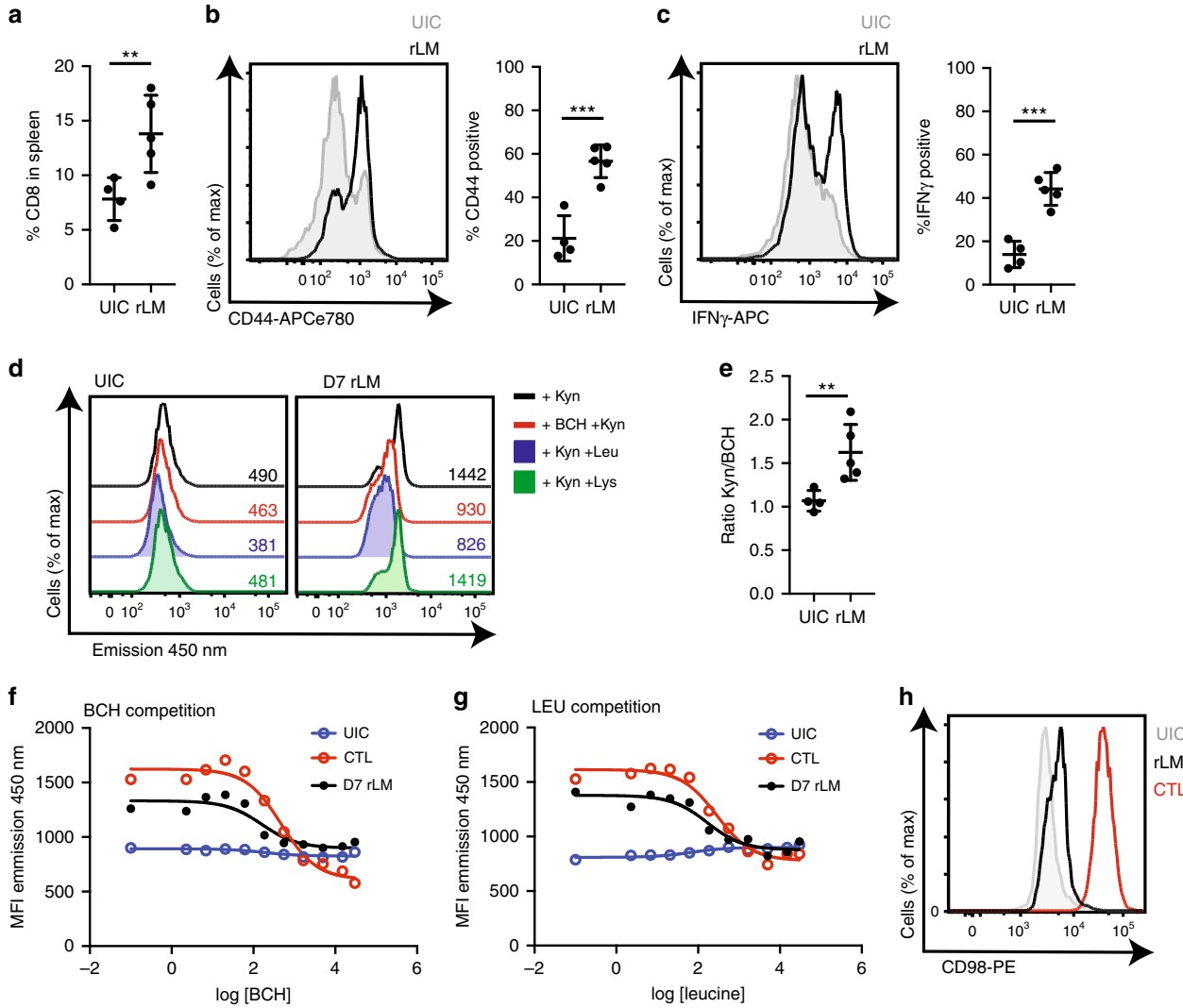

**Fig. 3** System L flow cytometry uptake assay with a mixed cell population. **a** Frequency of CD8⁺ T cells, **b** CD44 and **c** IFNγ expression in CD8⁺ T cells from spleen 7 days after recombinant *Listeria monocytogenes* (rLM) infection compared with uninfected controls (UIC). **d** 450 nm fluorescence emission of CD8⁺ T cells from uninfected controls (UIC; left) or D7 rLM (right) treated with kynurenine in the presence or absence of BCH (10 mM), Leu (5 mM) or Lys (5 mM) at (37 °C, 4 min). **e** Ratio of the 450 nm emission of CD8⁺ T cells treated with kynurenine ± BCH 10 mM (4 min, 37 °C). **f, g** Mean 450 nm fluorescence emission of CD8⁺ T cells from UIC, D7 rLM or CTL treated with kynurenine in the presence of increasing concentrations of BCH (**f**) or leucine (**g**) at (37 °C, 4 min). **h** CD98 surface expression of CD8⁺ T cells from UIC, D7 rLM or CTL. Error bars are ±s.d. from five biological replicates. Points indicate biological replicates (**a**–**c**, **e**). MFIs are indicated in the histograms (**d**), data are representative of five biological replicates (**d**, **f**–**h**). *P* values * = < 0.01; ** = < 0.005; *** = < 0.001; **** = < 0.0001 (unpaired *t*-test)

T cells increase System L transport compared with uninfected controls (UIC) (Fig. 3e). To verify the specificity of the observed increase in kynurenine transport seen in the in vivo generated CD8⁺ effector T cells, we performed the uptake assay in the presence of increasing concentrations of the System L transporter inhibitor, BCH or in the presence of increasing concentrations of the System L preferred substrate, leucine. The data in Fig. 3f and g show clearly that kynurenine uptake by in vivo generated CD8⁺ effector T cells is competed by both BCH and leucine in a dose dependent manner. Furthermore, this is comparable to the competition data obtained from in vitro generated effector CTLs. The calculated IC50 values of BCH and leucine on kynurenine transport in D7 rLM CD8⁺ effector T cells are 161 and 168 µM respectively, whereas the IC50 values of BCH and leucine on kynurenine transport in CTLs are 492 µM and 411 µM respectively (Table 2). The concordance between the IC50s of BCH and leucine values is in agreement with kynurenine being transported by a System L transporter, and the higher IC50 values seen in the

CTLs would suggest these in vitro derived effectors express higher levels of System L transporters than in vivo generated effector CD8⁺ T cells. In support of this, CD98 expression (a.k.a SLC3A2; heavy chain subunit of heterodimeric System L transporters) is higher on in vitro maintained CTLs compared to D7 rLM CD8⁺ T cells (Fig. 3h).

Collectively, these data show that T cells activated in vitro and in vivo upregulate kynurenine transport capacity through a BCH-sensitive, System L amino acid transporter. They also demonstrate that the spectral properties of kynurenine can be exploited to allow analysis of kynurenine transport in individual cells within tissues and mixed cell populations.

**SLC7A5 is the dominant kynurenine transporter in T cells**. The BCH sensitivity and sodium independence of kynurenine transport in T cells indicates that it is transported by System L

**Table 2 IC50 values of BCH and leucine on kynurenine uptake**

|       | BCH      | LEU      |
|-------|----------|----------|
| CTL   | 492 ± 45 | 411 ± 30 |
| D7 rLM| 161 ± 34 | 168 ± 38 |

The IC50 (µM) values of BCH or LEU on kynurenine uptake in CD8[+] T cells from spleen 7 days after recombinant *Listeria monocytogenes* (D7 rLM) infection and CTL. IC50 values were calculated from the dose–response curves (Fig. 3f, g) using one-site Fit IC50, GraphPad Prism. Data are from five biological replicates

amino acid transporters. These are heterodimers composed of a heavy chain chaperone unit, CD98 (SLC3A2) and a light chain responsible for amino acid transport, e.g. SLC7A5 (LAT1) and SLC7A8 (LAT2)[22]. We have previously identified that the dominant System L transporter expressed in activated CD8[+] T cells is SLC7A5[24]. To probe the importance of SLC7A5 expression for kynurenine uptake, we used CD4-Cre *Slc7a5*[fl/fl] mice, which express Cre recombinase under the control of the CD4 promoter, thus driving deletion of the *Slc7a5* gene in CD4[+]CD8[+] double-positive (DP) thymocytes and all subsequent T-cell populations[24]. The data in Fig. 4a show that SLC7A5 null CD4[+] T cells do not take up radiolabelled kynurenine in response to TCR activation. Together these data, and the data presented above, reveal that kynurenine transport in lymphocytes is tightly controlled and mediated predominantly through the System L amino acid transporter SLC7A5. This explains why kynurenine transport in T-cell populations is restricted to immune-activated T cells and not seen in naive T cells; this is the pattern of SLC7A5 expression in T cells[24, 25].

In further experiments, we set up co-cultures of wild-type (CD45.1) and CD4Cre *Slc7a5*[fl/fl] (CD45.2) splenocytes and activated the T cells with CD3/CD28 antibodies. Figure 4b shows representative gating of the CD45.1[+] (wild-type) and CD45.2[+] (CD4Cre *Slc7a5*[fl/fl]), CD4[+] and CD8[+] subpopulations within the splenocyte mix. The data in Fig. 4c and d compare kynurenine uptake from co-cultured wild-type and CD4Cre *Slc7a5*[fl/fl] CD4[+] and CD8[+] T cells, respectively. Kynurenine uptake is increased in wild-type T cells in response to TCR stimulation, but does not increase in the SLC7A5 null T cells. The System y + L preferred substrate, lysine, did not inhibit kynurenine uptake in the activated wild-type T cells, whereas BCH treatment did, thus confirming that the kynurenine transport, as measured by flow cytometry, was via System L transporters (Fig. 4d–f).

**System L dependence of kynurenine activation of the AHR.** Kynurenine has been reported to act as an endogenous ligand for the AHR[2, 15, 18, 26]. To assess the importance of System L-mediated transport of kynurenine for its ability to act as an AHR ligand, we explored the capacity of CD4[+] T cells to transport kynurenine as they polarise and differentiate to the TH17 lineage. CD4[+] TH17 cells express high levels of the AHR and show strong responses to AHR ligands[13, 26]. In these experiments, CD4[+] T cells were polyclonally activated with CD3/CD28 antibodies and cultured with the cytokines IL-6, IL-1β and TGFβ (TH17[cond]). Figure 5a shows that uptake of [3]H-Kyn is low in the naive CD4[+] T cells and strongly upregulated in day 2 and day 3 activated CD4[+] T cells, but by day 5 [3]H-Kyn uptake has returned to near baseline low levels. The [3]H-Kyn uptake by the activated CD4[+] T cells was always dependent upon BCH-sensitive System L transport (Fig. 5a). We also used the flow cytometry-based assay to monitor kynurenine uptake and found that CD4[+] T cells activated under TH17 polarising conditions show high levels of

kynurenine uptake on day 3 of culture but this declines at day 5 (Fig. 5b, c).

Does System L transport control kynurenine-mediated AHR signalling? To investigate this, we monitored the ability of kynurenine to induce the expression of mRNA encoding the AHR target, *Cyp1A1* in CD4[+] TH17[cond] cells on D3, with high levels of SLC7A5-mediated transport, compared to D5 post activation, where cells have reduced levels of SLC7A5-mediated kynurenine transport. As a control, we used the AHR ligand 6-formylindolo[3,2-b]carbazole (FICZ), a tryptophan-derived photoproduct. The intrinsic fluorescence of FICZ has previously been used to monitor cellular accumulation of FICZ by flow cytometry using BP filter for 530/30 nm emission with 488 nm laser excitation[27]. Figure 5d shows that T cells rapidly take up FICZ but this is not affected by System L inhibition with BCH, whereas kynurenine uptake is inhibited by BCH (Fig. 5d–f). Importantly, the data in Fig. 5g show that the ability of kynurenine to induce *Cyp1A1* mRNA expression in TH17 polarised CD4[+] T cells was strikingly higher on day 3 post activation compared to day 5 cells, which correlates with the relative ability of the cells to transport kynurenine (Fig. 5a, b). This is in contrast to FICZ mediated *Cyp1a1* mRNA expression, which is equivalent on D3 and D5 (Fig. 5g). Further evidence for the importance of System L transport for kynurenine action came from experiments using BCH treatment to inhibit System L transport. The data in Fig. 5h show that BCH impedes *Cyp1a1* mRNA induction in response to kynurenine, but does not block FICZ mediated *Cyp1a1* mRNA induction. To further investigate the parameters of kynurenine-mediated AHR signalling on CD4[+] TH17[cond] cells, we performed a dose–response of kynurenine and measured corresponding *Cyp1A1* mRNA expression in D3 TH17[cond] cells. The EC50 of kynurenine to drive *Cyp1A1* mRNA expression was 11.07 µM (Fig. 5i). This correlates tightly with the EC50 for AHR activation by kynurenine in COS-1 cells (13 µM)[16], and the EC50 of kynurenine to drive *Cyp1A1* expression in U87 glioma cells (12.3 µM)[2]. Moreover, the data in Fig. 5j show that induction of *Cyp1A1* mRNA in D3 TH17[cond] cells in response to kynurenine (10 µM) is dependent on AHR signalling, as it is completely blocked in the presence of an AHR antagonist. These data collectively show that kynurenine can activate AHR signalling in activated CD4[+] T cells, but only when cells express a System L transporter. Furthermore, the blockade of kynurenine transport with System L inhibitors blocks kynurenine-mediated AHR signalling.

**Discussion**

The present data show that kynurenine uptake in T cells is mediated by the System L amino acid transporter SLC7A5. The expression of SLC7A5 is precisely regulated by immune activation[24] and, accordingly, the ability of T cells to take up kynurenine is restricted to immune-activated T cells. The importance of this regulated transport of kynurenine is illustrated by the fact that the ability of kynurenine to function as a ligand for the AHR is limited by the System L transport capacity of the T cell. Hence, only cells expressing high levels of System L transporter activate the AHR in response to kynurenine; blocking System L transport inhibits AHR activation in response to kynurenine. This could explain why relatively high extracellular concentrations (µM) levels of kynurenine are needed to act as an AHR ligand compared with the effects of other AHR ligands (e.g. TCCD or FICZ) that are proposed to passively diffuse across the plasma membrane. In this respect, recent work has reported trace element condensation products of kynurenine to be a potent AHR ligand[16]. Our data showing the System L dependence of kynurenine activation of the AHR indicate that this trace metabolite may also need to be actively transported

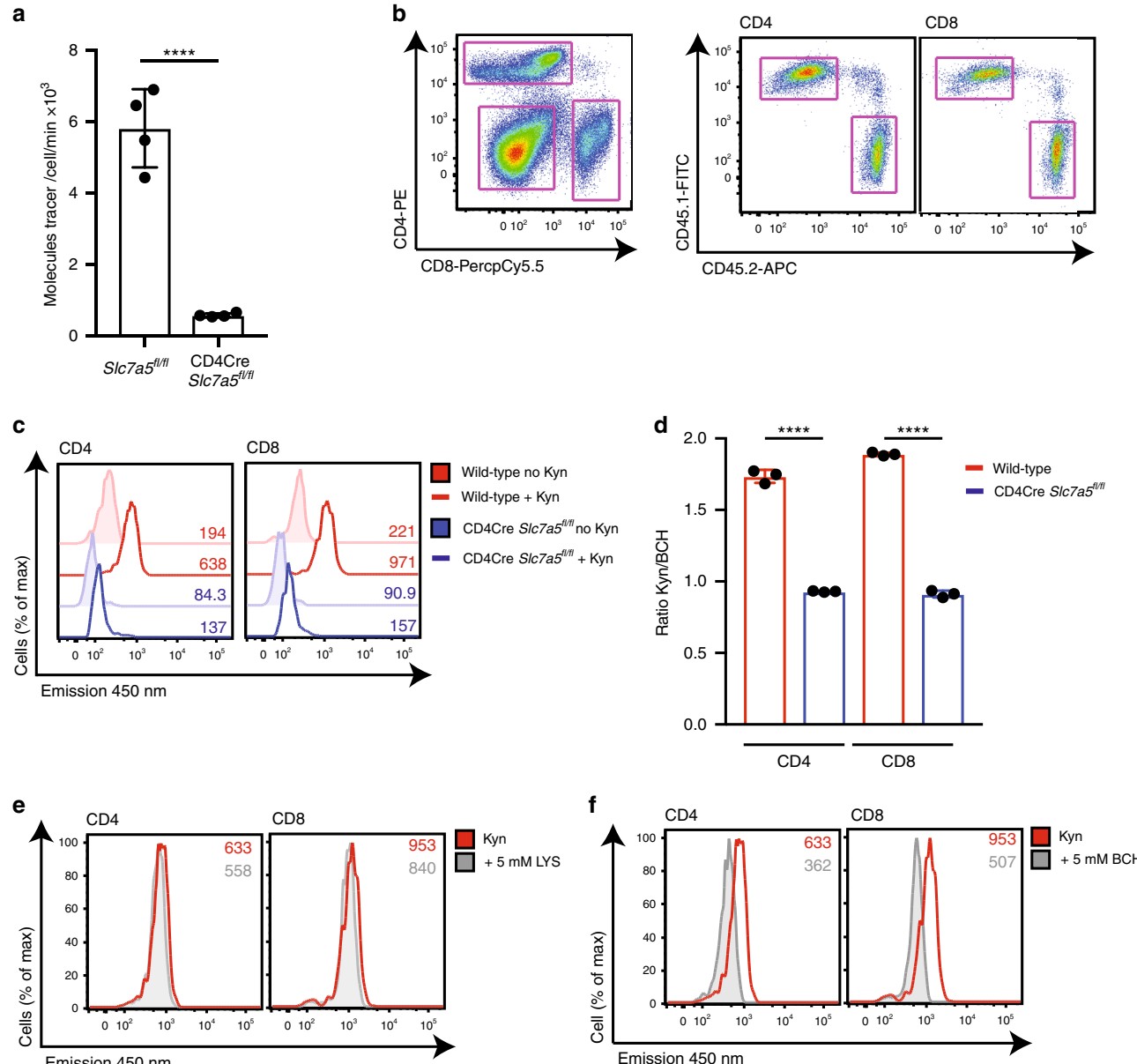

**Fig. 4** SLC7A5 mediates kynurenine transport in activated T cells. **a** Uptake of $^3$H-kynurenine in TCR-stimulated (CD3/CD28, 18 h) CD4$^+$ T cells from *Slc7a5$^{fl/fl}$* or CD4Cre$^+$ *Slc7a5$^{fl/fl}$* mice. **b–d** Wild-type (CD45.1) and CD4Cre$^+$ *Slc7a5$^{fl/fl}$* (CD45.2) splenocytes were mixed 1:1 and activated using CD3/CD28 antibodies for 24 h. **b** The data show representative flow cytometry profiles of the gating strategy used to identify CD4$^+$, CD8$^+$ (left panel) and subsequent CD45.1$^+$ and CD45.2$^+$ subpopulations (right panel). **c** The overlaid histograms show 450 nm emission in wild-type and CD4Cre$^+$ *Slc7a5$^{fl/fl}$* CD4$^+$ (left panel) or CD8$^+$ (right panel) T cells after 4 min (37 °C) with or without kynurenine (200 μM). MFIs are indicated. **d** Ratio of the 450 nm emission of T cells treated with kynurenine ± BCH 10 mM. **e**, **f** Histograms showing 450 nm fluorescence emission of wild-type CD4$^+$ (left panel) and CD8$^+$ (right panel) T cells after 4 min kynurenine (200 μM, 37 °C) uptake in the presence or absence of 5 mM lysine, a System y + L competitor (**e**); or 5 mM BCH, a System L inhibitor (**f**). Error bars are ±s.d. Individual points indicate biological replicates. MFIs are indicated in the histograms, data are representative of four (**a**) or three (**b–f**) biological replicates. *P* values **** = < 0.0001 (unpaired *t*-test, **a**; ordinary one-way ANOVA, **d**)

into the cell. The ability of kynurenine to enter cells is thus dictated by the Km of System L transporters for kynurenine. In this respect, studies that consider how kynurenine controls the immune system, be it AHR-dependent or independent, do not consider that cells will not be capable of sensing intracellular kynurenine unless they have been activated to express System L transporters.

The knowledge that kynurenine is a System L substrate, and more specifically an SLC7A5 substrate, also affords new insight

about why kynurenine might be immunosuppressive. SLC7A5 substrates thus include other indispensable large neutral amino acids such as leucine and hence SLC7A5 is required for the activation of the leucine sensing serine kinase pathways mediated by mammalian target of rapamycin complex1 (mTORC1)[24]. SLC7A5-mediated amino acid transport is also critical for maintaining expression of the transcription factor c-Myc, a key regulator of lymphocyte metabolism[25,28]. There may thus be multiple AHR independent immunosuppressive effects

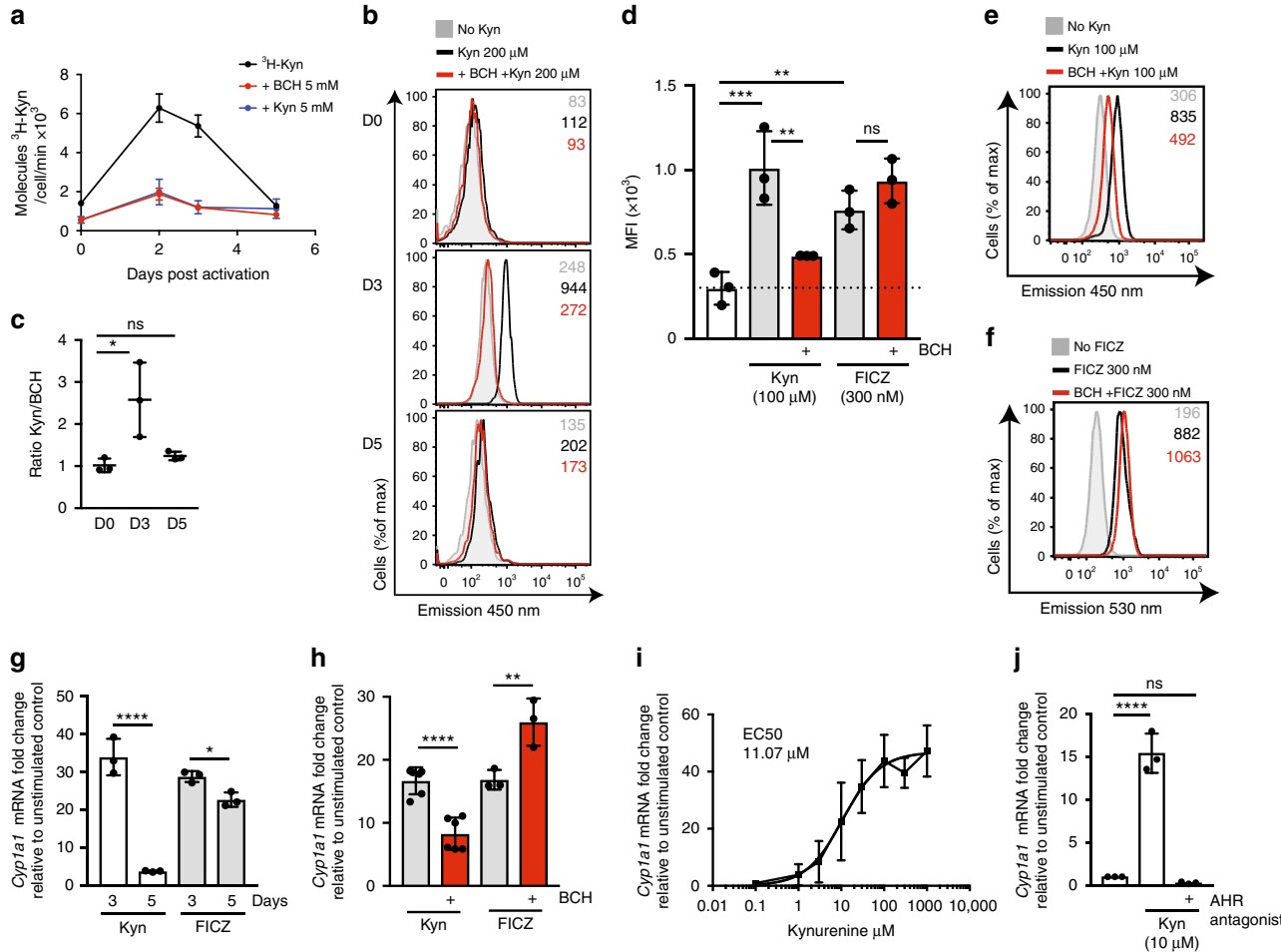

**Fig. 5** System L-mediated transport of kynurenine drives AHR signalling. **a** CD4$^+$ T cells were activated and maintained in TH17 conditions (CD3/CD28; IL-6, IL-1β and TGFb) (TH17$^{cond}$). Uptake of $^3$H-kynurenine in the presence or absence of BCH (5 mM) or non-radiolabelled kynurenine (5 mM) at times D2, D3 and D5 post activation is shown. **b** Histograms showing 450 nm fluorescence of D0, D3 and D5 TH17$^{cond}$ CD4$^+$ T cells treated with kynurenine (200 μM, 4 min) in the presence or absence of BCH (5 mM), or with no kynurenine. **c** Ratio of the 450 nm emission of D0, D3 and D5 TH17$^{cond}$ CD4$^+$ T cells treated with kynurenine ± BCH 5 mM (4 mins, 37 °C). **d** The graph shows the mean fluorescence intensity (MFI) of D3 TH17$^{cond}$ CD4$^+$ T cells treated with kynurenine (100 μM, 4 min, emission 450 nm) or FICZ (300 nM, 4 min, emission 530 nm) in the presence or absence of BCH (10 μM). **e**, **f** Representative flow cytometry histograms of (**d**); kynurenine uptake (**e**) and FICZ uptake (**f**). **g** The graph shows *Cyp1a1* mRNA from D3 or D5 TH17$^{cond}$ CD4$^+$ T cells stimulated with kynurenine (100 μM, 2 h) or FICZ (300 nM, 2 hr), relative to non-treated control. **h** Induction of *Cyp1a1* mRNA in D3 TH17$^{cond}$ CD4$^+$ T cells stimulated with kynurenine (100 μM, 2 h) or FICZ (300 nM, 2 h) in the presence or absence of BCH (25 mM). The data are shown relative to non-treated control. **i** Dose–response induction of *Cyp1a1* mRNA in D3 TH17$^{cond}$ CD4$^+$ T cells stimulated with kynurenine at indicated doses for 2 h. The data are shown relative to non-treated control. **j** Induction of *Cyp1a1* mRNA in D3 TH17$^{cond}$ CD4$^+$ T cells stimulated with kynurenine (10 μM, 2 h) in the presence or absence of AHR antagonist (CH223191, 30 μM). The data are shown relative to non-treated control. Error bars are ±s.d. Individual points indicate biological replicates. MFIs are indicated in the histograms, data are from three biological replicates (**a–g**, **i–j**) and six (Kyn) or three biological replicates (FICZ) (**h**). *P* values * = < 0.01; ** = < 0.005; *** = < 0.001; **** = < 0.0001; ns= not significant (ordinary one-way ANOVA)

of kynurenine regulated through amino-acid competition for uptake or exchange through SLC7A5. However, the ability of kynurenine to function as an immunomodulatory molecule via this 'competition model' or to activate the AHR will be determined by concentrations of other System L substrates in the T cell microenvironment. We show the Km of kynurenine transport in CTLs is ~200 μM, whereas the Km of other System L substrates (eg leucine, tryptophan or methionine) is ~10-fold less. Plasma levels of System L substrates range from 20–100 μM and are relatively constant[29, 30], whereas plasma levels of kynurenine are rarely above 3 μM and subject to efficient clearance[3, 31, 32]. Taken together, this indicates that the levels of kynurenine required to out-compete the much higher affinity substrates exceed the physiological kynurenine levels measured

to date, and allow us to conclude that only in microenvironments of fierce nutrient competition would this particular scenario ever be plausible. Do microenvironments exist where kynurenine could function as a AHR ligand? In this context, high concentrations of kynurenine (50–100 μM) are consistently used to demonstrate AHR activation in vitro[1, 2, 7, 14, 33]. These concentrations are far in excess of kynurenine plasma levels typically found in the literature (~1–3 μM)[3, 31, 32]. Unfortunately, there are very few studies that include any detailed, microenvironmental quantitative metabolite analysis. However, Opitz et al. measured kynurenine concentrations of 37 μM in a U87 xenograft tumour model. This is within the range of what could activate the AHR given low levels of competition with other SLC7A5 substrates. In this

context, it is beginning to be realised that the tumour micro-environment can be depleted of key nutrients. For example, Ho et al.[34] demonstrated a 10-fold reduction in glucose concentrations measured from tumours compared with matched spleen or blood samples. Using these measurements to extrapolate a 10-fold reduction of System L substrate amino acids and a 10-fold increase in kynurenine levels in IDO-expressing tumour environments at the same time, could create an environment whereby kynurenine levels would be significantly high enough to drive AHR activation as well as apply competitive pressure upon SLC7A5-mediated transport of essential amino acids. Hence, local/microenvironmental kynurenine levels would undoubtedly be critically important but even then, our data clearly demonstrate that only SLC7A5 expressing T cells will be capable of transporting kynurenine into the cell interior. The ability of kynurenine to act as an immunomodulatory metabolite would thus be limited.

Notwithstanding the question of the in vivo significance of kynurenine as an AHR ligand, one useful property of kynurenine that facilitated characterisation of its membrane transport is its spectral properties. These have permitted the development of a flow cytometry-based assay that allows rapid, sensitive and quantitative measurement of kynurenine transport in single cells. The FACS-based kynurenine uptake assay allows analysis of individual cell responses in tissues and provides a rapid assay to assess the capacity of T cells or other cells to be susceptible to the immunosuppressive effects of kynurenine. It is also highly pertinent that kynurenine is a System L amino acid transporter substrate in T cells; this knowledge allows the uptake of kynurenine to be used as an assay for cellular System L transport. In this respect, the current "gold standard" for investigating System L activity is to use radiolabelled amino acids that are System L substrates (e.g. leucine, phenylalanine) in short uptake assays, allowing transporter kinetics and biochemical parameters to be studied. However, radiolabelled amino acid uptake measures population responses and cannot interrogate nutrient transport in intricate cellular biological systems. This requires single cell-based assays such as the kynurenine flow cytometry-based assay described herein. The present data show that monitoring increases in cellular fluorescence in cells exposed to kynurenine allows quantitation of System L transporter activity in single lymphocytes and can therefore be used as a tool to measure System L amino acid uptake in rare cell populations in complex tissues. This is a very rapid and sensitive assay for System L transport. Importantly, the simplicity of the assay makes it easy to rigorously control for System L transport activity by assessing the sensitivity of the kynurenine uptake to the System L inhibitor BCH as well as its sensitivity to competition with System L substrates (e.g. leucine) versus non-System L substrates (e.g. lysine). This description of a single cell flow-based assay for amino acid transport is a novel and important advancement in a field which has previously been constrained to population-based approaches using radioligands or metabolite analyses.

## Methods

**Mice.** C57BL/6J (wild-type, WT), CD4Cre *Slc7a5*[fl/fl] and C57BL/6J Ly5.1 (CD45.1) mice were bred and maintained in the WTB/RUTG, University of Dundee. All studies were performed on project license PPL60/4488, approved by the University of Dundee Welfare and Ethical Use of Animals Committee and in compliance with UK Home Office Animals (Scientific Procedures) Act 1986 guidelines. Female mice aged 12–20 weeks were used.

**Cells.** To activate primary T cells, spleens and/or lymph nodes (LN) were removed, disaggregated and red blood cells lysed. Cells were stimulated with 1 µg/ml of the CD3 monoclonal antibody (2C11) and 2 µg/ml anti-CD28 (37.51; ebiosciences) to trigger the TCR. Naive CD4$^+$ T cells and 24 h TCR-activated

CD4$^+$ T cells were purified using CD4 isolation kit (EasySep, STEMCELL Technologies). To generate CTLs, cells were washed from TCR stimulus after 48 h and then expanded in IL-2 (20 ng/ml; Proleukin, Novartis) for a further 3–5 days. To generate TH1s and TH17s, murine CD8$^+$ T cells were depleted from lymph node preparations using CD8 depletion kit (EasySep, STEMCELL Technologies). The resulting mix of CD4$^+$ T cells and APC were cultured at $3 \times 10^5$ cells/ml for 5 days in the presence of anti-CD3 (2 µg/ml) and anti-CD28 (3 µg/ml) and, for TH1s; cytokines IL-12 (10 ng/ml; RnD Systems) and IL-2 (20 ng/ml) and for TH17s; cytokines IL-6 (50 ng/ml, Peprotech), IL-1β (10 ng/ml, Peprotech) and TGFβ (2 ng/ml, Peprotech). TH17s were cultured in IMDM (Gibco). All other cells were cultured in RPMI 1640 containing L-glutamine (Gibco). Culture media was supplemented with 10% FBS (Gibco), 50 µM β-mercaptoethanol (β-ME, Sigma) and penicillin/streptomycin (Gibco). Cells were incubated at 37 °C with 5% CO$_2$ throughout.

Where indicated, cells were treated with 6- formylindolo[3,2-b]carbazole (FICZ, Biomol, Enzo Life Sciences), kynurenine (Sigma), 2-amino-2-norbornanecarboxylic acid (BCH, Sigma) or the AHR antagonist CH223191 (Sigma) for the indicated concentrations and times.

**Radiolabelled tracer uptake.** Briefly, radiolabelled uptake was carried out using $1 \times 10^6$ cells resuspended in 0.4 ml uptake medium. Kynurenine uptake was carried out in warmed (37 °C) HBSS (GIBCO) containing [$^3$H] L-kynurenine (0.1 µCi/ml). 4 min uptake assays were carried out layered over 0.5 ml of 1:1 silicone oil (Dow Corning 550 (BDH silicone products; specific density, 1.07 g/ml):dibutyl phthalate (Fluka)). Cells were pelleted below the oil, the aqueous supernatant solution, followed by the silicon oil/dibutyl phthalate mixture was aspirated, and the cell pellet underneath resuspended in 200 µl NaOH (0.5 M) and β-radioactivity measured by liquid scintillation counting in a Beckman LS 6500 Multi-Purpose Scintillation Counter (Beckman Coulter). Similarly, glutamine uptake was performed using [$^3$H] L-glutamine (0.5 µCi/ml); tryptophan uptake was performed using [$^3$H] L-tryptophan (0.5 µCi/ml). Where indicated, 5 mM BCH, Kynurenine, L-glutamine or L-tryptophan were used, respectively, to quench radiolabeled ligand uptake. Where indicated, the sodium free buffer was prepared using TMACl as described in Baird et al.[35] Data are expressed as molecules radiotracer per cell per minute. [$^3$H] L-kynurenine, [$^3$H] L-tryptophan and [$^3$H] L-glutamine were obtained from Perkin Elmer. All other chemicals were obtained from Sigma.

**Flow cytometry.** For cell surface staining, antibodies conjugated to FITC, PE, APC, Alexafluor 647 (A647), APC-efluor780 (APCe780), and PerCPCy5.5 were obtained from either BD Pharmingen, eBioscience or Biolegend. Fc receptors were blocked using Fc Block (BD Pharmingen). Antibody clones used were: CD4 (RM4-5), CD8a (53-6.7), TCRβ (H57-597), CD44 (IM7), CD45.1 (A20), CD45.2 (104) and CD98 (RC388). Cells were fixed using 1% paraformaldehyde. Standard intracellular cytokine staining protocols were followed for IFNγ (clone XMG1.2; Biolegend) staining. Data were acquired on a LSR Fortessa II with DIVA software or a FACSVerse flow cytometer with FACSuite software (BD Biosciences) and analysed using FlowJo software version 9.9.5 (TreeStar).

**Single cell assay to monitor System L amino acid transport.** Start with at least $1 \times 10^6$ cells per condition, to obtain at least 200,000 events analysed by flow cytometry at the end of the protocol. Conditions should include: the test samples (treated with kynurenine); background fluorescence control samples (matched samples, not treated with kynurenine, thus allowing for identification of cells exhibiting fluorescence above background); specificity controls such as System L blocked samples (BCH-treated, or Leu-treated), or uptake performed on ice (4 °C) (to determine transported kynurenine as opposed to surface binding); positive controls such as cell treatments driving high expression of System L transport. If required, surface cell antibody staining should be performed prior to uptake assay protocol. This may be performed at room temperature or 4 °C, however test samples must be warmed to 37 °C, e.g. in water bath, prior to kynurenine uptake. (As with all multi-parameter flow cytometry, appropriate antibody staining controls must also be performed.) Samples can be fixed immediately after uptake assay by addition of 4% (vol/vol) paraformaldehyde (PFA; to a final concentration of 1%) for 30 min at room temperature.

For kynurenine uptake assay; pre-warm kynurenine (800 µM, in HBSS), BCH (40 mM, in HBSS) and lysine (20 mM, HBSS) and HBSS to 37 °C. After surface antibody staining of samples, resuspend cells in 200 µl warmed HBSS ($1–5 \times 10^6$ cells in FACS tubes, or scale accordingly into plates). Keep cells in water bath at 37 °C. Add 100 µl of HBSS, or BCH or lysine to appropriate samples. Add 100 µl HBSS to no kynurenine controls (final volume 400 µl). Finally, add 100 µl kynurenine to appropriate samples. Stop uptake after 4 min by adding 125 µl 4% PFA for 30 min at room temperature, in the dark. The final concentrations for uptake assay are: 200 µM kynurenine; +/-10 mM BCH; +/-5 mM lysine. After fixation, wash cells twice in PBS/0.5% BSA and resuspend in PBS/0.5% BSA prior to acquisition on flow cytometer. The 405 nm laser and 450/50 BP filter are used for kynurenine fluorescence detection.

To monitor kynurenine uptake in live cells, acquire data on flow cytometer immediately following addition of kynurenine and plot fluorescence against time.

**Recombinant *Listeria monocytogenes* infection**. Mice were infected with attenuated ActA-deleted *L. monocytogenes* i.v. with $10 \times 10^6$ colony forming units[36], and T lymphocytes from spleens were analysed by flow cytometry on D7 post-infection.

**Quantitative real-time PCR**. RNA was purified using the RNeasy RNA purification Mini Kit (Qiagen) (Genomic DNA was digested with RNase-free DNase (Qiagen) following manufacturer instructions) and reverse-transcribed using the iScript cDNA synthesis kit (BioRad). Quantitative PCR was performed in 96-well plate format using iQ SYBR Green-based detection (BioRad) on a BioRad iCycler.
Cyp1A1 qPCR primers:
Forward: TGCCTAACTCTTCCCTGGATGCCTT
Reverse: CCGGATGTGGCCCTTCTCAAATGT

**Statistics**. Group/sample sizes for individual experiments or animal studies were determined by previous pilot studies and experience with regard to the experimental parameters measured. No samples/animals were excluded, neither blinding nor randomisation was used. All data analysis was performed using GraphPad Prism version 7.0, GraphPad Software. A Shapiro–Wilkes normality test was performed followed by either unpaired $t$-tests or one-way ANOVA with Dunnett's multiple comparison test. Variance was similar between the groups that were statistically compared. $P$ values $* = < 0.01$; $** = < 0.005$; $*** = < 0.001$; $**** = < 0.0001$; ns = not significant. Specific tests used are stated in the figure legends.

**Adoptive transfer and ova immunisation**. For in vivo activation OT2 (CD45.1) lymph node cells were transferred into C57/Bl6 (CD45.2) hosts. After 24 h, mice were immunised i.p. with 4-Hydroxy-3-nitrophenylacetyl hapten conjugated to ovalbumin (NP-OVA; 100 µg; BioSearch technologies) adsorbed to alum (Pierce). Spleens were harvested and transferred cells were identified and analysed for activation and proliferation at D0, D1 and D3 after immunisation, respectively.

**Data availability**. The data that support the findings of this study are available from the corresponding author on reasonable request.

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

## Acknowledgements

We thank Cantrell group members for their critical discussion of the data, the Biological Resources unit, Sarah Thomson (for rLM work) and the Flow Cytometry facility (A. Whigham and R. Clarke) at the University of Dundee. This work was supported by the Wellcome Trust (Principal Research Fellowship to D.A.C. 097418/Z/11/Z and 205023/Z/16/Z, and Wellcome Trust Equipment Award 202950/Z/16/Z).

## Author contributions

L.V.S., G.R. and D.A.C. conceptualised the project; L.V.S. and D.N. performed the experiments; L.V.S. and D.A.C wrote the original draft of the manuscript; L.V.S., D.N., G. R., P.M.T. and D.A.C. reviewed and edited the manuscript. L.V.S. and D.A.C. supervised the research.

## Additional information

Competing interestThe authors declare no competing interests.

