## [Peer Review File · Nature Communications]

Reviewers' comments:

Reviewer #1 (Remarks to the Author):

This is an interesting, well written study demonstrating the requirement for System L amino acid transport for uptake of kynurenine by T cells.

Kynurenine, the major tryptophan metabolite generated by the IDO pathway has attracted a lot of attention in the immune community for its potent suppressive effects. The data presented in this manuscript convincingly show that kynurenine needs to be transported into cells and identify the relevant transporter. Furthermore, the data also clearly show the requirement for activation to upregulate the transporter. Less convincing are the data on AhR activation by kynurenine.

Specific points:

Suppl. Fig. 1A should be included in the main manuscript as the temperature dependency is a very important technical detail

It would be important to measure the expression of transporter by qPCR in CTL generated in vitro vs those isolated after *Listeria* infection rather than just 'suggest' the levels are higher on in vitro generated cells. What aspect of activation correlates with expression of the transporter?

The data suggesting that AhR signalling via kynurenine is controlled by System L transport in Fig. 5 are less convincing. One question arising is how kynurenine gets out of an IDO expressing cell to access Th17 cells as these do not express IDO? Does this involve transport again?

The concentrations used to visualise AhR pathway activation are extremely high (as they are in the literature when kynurenine is proposed as AhR ligand). The authors argue that kynurenine transport only functions efficiently if a cell is activated to express the transporter. However, if 100µM kynurenine is required to activate AhR even in an optimally activated cell this begs the question whether this can be regarded as physiologically relevant.

Another concern is the readout of AhR activation at static time points of 3 or 5 days. Cyp1a1 expression is very transient under physiological conditions and the kinetics very much depend on how quickly ligands are metabolised. FICZ on the other hand is a strong activator and the doses used in these cultures are exceptionally high so that despite rapid metabolism via Cyp1a1 there is probably sufficient FICZ available to sustain activation. Rather than measuring Cyp1a1 it would be better to measure the physiological consequence of AhR stimulation in Th17 cells, which is production of IL-22.

Minor points:

Some of the citations are not correct. Ref. 1 and 2 have nothing to do with AhR promoting T cell differentiation and Ref. 25 does not deal with kynurenine as AhR ligand

Why are the concentrations of BCH inhibitor used varied from 5mM to 25nM?

Reviewer #2 (Remarks to the Author):

In "Single cell analysis of Kynurenine and System L amino acid transport in T lymphocytes" the authors present a well-written study clearly describing a novel method for monitoring kynurenine uptake dependent on System L transporter SLC7A5 by flow cytometry for single cell analysis of T lymphocytes. Ultimately, the novelty of the paper appears to this reviewer to lie primarily in the methods and less so in the biologic application of this method. The data showing the role of

System L for Kynurenine uptake and the use of the natural fluorescence of kynurenine are convincing and clear. There are a number of weaknesses that should be addressed to better validate the assay for endogenous levels of kynurenine and demonstrate the effects are functionally important.

A key goal and benefit of the fluorescence assay is the ability to analyze kynurenine in mixed cell populations and even endogenous kynurenine in vivo. The paper, however, only uses high levels of kynurenine on T cells treated in vitro. It would be very important to see if this method can detect endogenous levels in vivo. For example, transfer of T cells to a tumor model in which tumor cells are IDO1 expressing or not, or +/- IDO1 inhibitor would be useful to look at the impact on T cell kynurenine uptake.

Given the role of AhR in Th17 and Treg balance, it would be interesting to also look at Treg and show how kynurenine uptake may be regulated in this subset. Also, Th17 cell uptake of kynurenine is measured, but does this correlate with IL17 production? Based on literature, AhR should decrease Th17 cells.

Statistics throughout should be evaluated to provide indications of significance. Error bars are provided, but statistical tests and indications of significance are not.

Reviewer #3 (Remarks to the Author):

This manuscript describes how activated T cells import the tryptophan catabolite kynurenine (Kyn) using a novel flow cytometric method to detect Kyn in individual T cells. The pretext is that Kyn was reported previously to regulate T cell responses by activating the aryl hydrocarbon receptor (AhR). This notion has been controversial because Kyn is a relatively weak AhR ligand. The major claims are that activated T cells import Kyn actively via the System L amino acid transporter SLC7A5 to incite AhR signaling. These new observations suggest that Kyn serves as a natural T cell suppressant by targeting activated T cells selectively, as previous work by the same group has shown that SLC7A5 is up-regulated only when T cells activate. Data presented supports these points and documents that the novel FACS-based assay described is a useful and robust method to detect Kyn uptake by activated T cells. This point notwithstanding, the study is strong on methodology but weak on biological significance, as key implications arising are not addressed in depth. Hence, appropriate attention to the points below has the potential to increase the scope and impact of this study.

Point 1:

Data shown in the right panel of Figure 2A suggest that all CTLs imported Kyn, Are System L (SLC7A5) transporters expressed uniformly by all cells in the 'CTL' population? Also, is Kyn uptake selective for CTLs when mixed with naïve (resting) T cells before adding Kyn?

Point 2:

In Figure 2, the authors cite KM values for Kyn uptake in the uM range but do not comment if this is likely to be in the same range as physiologic levels of Kyn. The authors refer to this issue in the opening paragraph of their Discussion but do not address it. This is a key point, as it relates to the current controversy regarding the significance of AhR signaling mediated by Kyn in physiologic settings.

Point 3:

In Figure 3, the authors show convincingly that all (CD8+) CTLs from mice infected with listeria (Lm) can import Kyn via System L. However, data presented raises additional questions regarding this model. First, did CTLs from Lm-infected mice show evidence of Kyn uptake in the absence of added Kyn? Shaded histograms shown in Fig 3B provide a hint that this may be the case. Linked to

this question, do CTLs import Kyn selectively in vivo, either due to IDO activity induced during Lm-infection or if Kyn is injected into Lm-infected mice? Second, did CD4 T cells (Foxp3-) and Tregs (Foxp3+) from Lm-infected mice also import Kyn? These points are of considerable interest because previous reports showed that Lm-infections induce IDO activity and that AhR signaling acts on Tregs to promote their regulatory functions.

Point 4:

Data presented in Figures 4 & 5 show convincingly that SLC7A5 ablation in CD4 T cells abolishes Kyn uptake in CD4 T cells selectively and that imported Kyn triggers AhR signaling in CD4 T cells, respectively. However, no data is presented to address the biological significance of these findings, specifically regarding the reported ability of Kyn to suppress Th1 responses (e.g. IFN γ production) or to reinforce Treg-mediated suppression of effector T cells via AhR signaling. As the pretext for this study was to address how Kyn activates AhR signaling to suppress T cell responses despite being a weak AhR ligand, data addressing these points would enhance the biological significance of this study.

Point 5:

In the Discussion, the authors speculate that Kyn may mediate suppression via AhR-independent mechanisms such as attenuating mTOR activation by limiting T cell access to essential amino acids. Is there any evidence to support these speculations?

Reviewer #1 (Remarks to the Author): (in black)

This is an interesting, well written study demonstrating the requirement for System L amino acid transport for uptake of kynurenine by T cells.

Kynurenine, the major tryptophan metabolite generated by the IDO pathway has attracted a lot of attention in the immune community for its potent suppressive effects. The data presented in this manuscript convincingly show that kynurenine needs to be transported into cells and identify the relevant transporter. Furthermore, the data also clearly show the requirement for activation to upregulate the transporter. Less convincing are the data on AhR activation by kynurenine.

Response to reviewer 1 (in red)

We would like to thank the reviewer for the many positive and insightful comments about our paper. We have revised the paper to address the issues raised. We understand the reviewer to be concerned regarding the role of kynurenine as an AhR ligand. We appreciate this apprehension and indeed one key aspect of our paper was to reveal that unlike conventional AhR ligands such as FICZ or TCDD, the ability of kynurenine to act as an AhR ligand will be limited to cells that can transport kynurenine.

Moreover, because kynurenine is transported by System L transporters the capacity of a cell to transport kynurenine will be limited by competition with other System L substrates. We have modified the discussion to make these points. We feel that too many groups just treat cells with kynurenine without any thought that it needs to be actively transported into cells. We hope our paper will raise awareness of this key fact.

Specific points:

Suppl. Fig.1A should be included in the main manuscript as the temperature dependency is a very important technical detail.

- We have now included the supplementary Figure as Figure 2F and 2G in the main manuscript. We are in agreement with the reviewer that this is an important technical detail.

It would be important to measure the expression of transporter by qPCR in CTL generated *in vitro* vs those isolated after *Listeria* infection rather than just 'suggest' the levels are higher on *in vitro* generated cells. What aspect of activation correlates with expression of the transporter?

- We apologise, it was not our intention to suggest that transporter levels are always increased *in vitro* compared to *in vivo*. Instead we meant to apply this statement specifically to the data that we showed. However, further to this, the reviewer raises a very interesting point regarding regulation of transporter expression. Data from the Immgen consortium show that mRNA levels of both CD98 (a.k.a SLC3A2; heavy chain subunit of the heterodimeric System L transporters) and SLC7A5 (light chain amino acid transporting subunit) transiently increase in CD8 T cells upon *in vivo* *Listeria* infection, peaking at 12-48h and declining by D6 post infection.

- In our previous studies we demonstrated that SLC7A5 expression is tightly regulated by TCR signaling and cytokines, in particularly gamma chain (IL-2) signaling. Thus, transporter levels are a kinetic response, dependent upon TCR and cytokine signaling. In every cell we have looked at, be it *in vivo* or *in vitro*, expression is tightly regulated and transient, corresponding to either having had high levels of recent TCR signaling, or IL2/gamma chain signaling. e.g. TH17 *in vitro* conditions do not provide conditions whereby transporter expression is sustained, however cells cultured in IL2 (Th1, CTL) maintain SLC7A5 and CD98 expression. We include flow cytometry staining of CD98 on uninfected control, *in vivo* D7 CD8 and *in vitro* generated CTLs to illustrate the increase in expression (Fig 3I). However, the log scale increase in CD98 expression cannot be used as a direct indicator of SLC7A5 expression. We feel that the discrepancy between very high CD98 expression (log scale changes) and not-so-high amino acid transport (linear scale differences) highlights the fact that being part of the heterodimer that transports large neutral amino acids is not the sole function of CD98, also supported by the CD98 knockout studies by Ginsberg et al. (Indeed, SLC7A5 null T cells express high levels of CD98.) We wish to reiterate how this uptake assay can report on which cells have the capacity to take up nutrients through transport using a single cell platform, where previously only population based analysis were possible. We have expanded this aspect in the discussion.

The data suggesting that AhR signalling via kynurenine is controlled by System L transport in Fig.5 are less convincing. One question arising is how kynurenine gets out of an IDO expressing cell to access Th17 cells as these do not express IDO? Does this involve transport again?

- We thank the reviewer for raising this point. Indeed, this would involve transport of kynurenine out from the IDO expressing cell into the microenvironment. It is well established that many tumours and immune cells upregulate nutrient transporters, including System L transporters and LAT1. Kynurenine and its metabolites have been shown to be transported out of the cell in exchange for tryptophan through LAT1 (Kaper et al, PLoS Biol 2007). It is likely that other LAT1 substrates would also exchange for kynurenine (System L transporters are obligate anti-port transporters: one in - one out). The fact that people routinely measure plasma and serum levels of kynurenine show that it is exported by the IDO expressing cell, or hepatic TDO expressing cells.

We believe our data show that *in vitro* kynurenine, transported through SLC7A5, can activate the AhR. Specifically, kynurenine drives expression of the AhR target Cyp1A1 (which is blocked by an AhR antagonist – new figure 5H). This Cyp1A1 expression is dependent upon kynurenine transport through SLC7A5 as it is inhibited by treatment with the System L inhibitor BCH (Fig 5f). The role, or ability, of kynurenine to activate the AhR *in vivo* is “up in the air” and this would very much be determined in a fierce nutrient competition environment, and have added this important discussion point.

The concentrations used to visualise AhR pathway activation are extremely high (as they are in the literature when kynurenine is proposed as AhR ligand). The authors argue that kynurenine transport only functions efficiently if a cell is activated to express the transporter. However, if 100µM kynurenine is required to activate AhR even in an optimally activated cell this begs the question whether this can be regarded as physiologically relevant.

- We fully agree with the reviewer that this is a very important and pertinent point. To address this we have included new data; a dose response of kynurenine activation

driving Cyp1A1 mRNA expression in D3 TH17 cells (maximal transporter expression, Fig 5G), and induction of Cyp1A1 mRNA in response to kynurenine (at 10uM) in the presence or absence of an AhR antagonist (Fig 5H). We hope this data will support our conclusion that kynurenine is an AhR ligand, capable of driving AhR signaling at lower concentrations than previously used in the literature – with the caveat that the transporter is expressed on the cells responding to kynurenine.

Interestingly, Seok et al have recently demonstrated that kynurenine itself acts more like a pro-ligand, requiring further “activation” for AhR signaling function; indeed the authors show that trace element condensation products of kynurenine (which they call TEACOPs; trace extended aromatic condensation products) are potent AhR agonists (in the picomolar range) (Seok et al, J Biol Chem, 2017). Our data does not disagree with this deeper/detailed study on the abilities of kynurenine as an AhR ligand indeed we feel that this further highlights the importance for T cells to transport the “pro-ligand” kynurenine into the cell.

We have expanded our discussion to address the physiological relevance of kynurenine as an AhR ligand.

Another concern is the readout of AhR activation at static time points of 3 or 5 days. Cyp1a1 expression is very transient under physiological conditions and the kinetics very much depend on how quickly ligands are metabolised. FICZ on the other hand is a strong activator and the doses used in these cultures are exceptionally high so that despite rapid metabolism via Cyp1a1 there is probably sufficient FICZ available to sustain activation. Rather than measuring Cyp1a1 it would be better to measure the physiological consequence of AhR stimulation in Th17 cells, which is production of IL-22.

- We apologise for this misunderstanding. The AhR activation conditions were not monitoring Cyp1a1 expression on D3 and D5 after long-term culture with kynurenine or FICZ. Rather, the cells were treated on D3 or on D5 post TCR activation; ie on those days cells were stimulated with the ligands (FICZ or Kyn) for a short pulse of 2 hrs. We feel that this data strongly supports our conclusions that kynurenine is only going to be a ligand for AhR in cells expressing transporter, and that as transporter expression is transient, so too is AhR responsiveness to kynurenine.
- The reviewer commented that it might have been valuable to look at kynurenine induction of IL-22 as this is a physiological consequence of AhR stimulation in Th17 cells. In this respect we have not measured the effect of kynurenine on IL-22 production. We agree that this is a very interesting target of the AhR and indeed in another study not yet published we have mapped all the the AhR/ARNT regulated genes in Th17 cells and find that there are many AhR/ARNT stimulated and repressed genes. Cyp1A is the most well documented AhR target and we felt that the detailed analysis of how kynurenine controls expression of Cyp1A makes the point that kynurenine can be an AhR ligand. Our study was to explore how kynurenine enters cells and was not intended to be a full study of the the biological consequences of a cells exposure to kynurenine. As we discuss in detail. We feel the fact that kynurenine needs to be transported by SLC7A5 raises many issues about when kynurenine would function as an AhR stimulus.

Minor points:

Some of the citations are not correct. Ref.1 and 2 have nothing to do with AhR promoting T cell differentiation and Ref.25 does not deal with kynurenine as AhR ligand

- We apologise for this mistake, and thank the reviewer for pointing it out. We have corrected this in the text.

Why are the concentrations of BCH inhibitor used varied from 5mM to 25nM?

- We understand the reviewer meant 25mM, as we do not use BCH at 25nM in this paper.
- BCH is a competitive blocker of the transporter LAT1 (SLC7A5/SLC3A2). As such, 5mM excess is sufficient for short uptake assay blockade (with no other competing substrates). Whereas 25mM is required to effectively maintain transporter blocking over prolonged cultures in the presence of other competing nutrients.

Referee #2 :

Reviewer #2 (Remarks to the Author): (in black)

In “Single cell analysis of Kynurenine and System L amino acid transport in T lymphocytes” the authors present a well-written study clearly describing a novel method for monitoring kynurenine uptake dependent on System L transporter SLC7A5 by flow cytometry for single cell analysis of T lymphocytes. Ultimately, the novelty of the paper appears to this reviewer to lie primarily in the methods and less so in the biologic application of this method. The data showing the role of System L for Kynurenine uptake and the use of the natural fluorescence of kynurenine are convincing and clear. There are a number of weaknesses that should be addressed to better validate the assay for endogenous levels of kynurenine and demonstrate the effects are functionally important.

Response to reviewer 2: (in red)

We would like to thank this reviewer for their positive comments on the paper. We feel so many groups appear to treat cells with kynurenine without ever considering that kynurenine must be actively transported into a cell. This is where the novelty of our paper lies as we hope people will read and start to consider this aspect of kynurenine biology.

We have responded to the specific issues raised by the reviewer below.

A key goal and benefit of the fluorescence assay is the ability to analyze kynurenine in mixed cell populations and even endogenous kynurenine *in vivo*. The paper, however, only uses high levels of kynurenine on T cells treated *in vitro*. It would be very important to see if this method can detect endogenous levels *in vivo*. For example, transfer of T cells to a tumor model in which tumor cells are IDO1 expressing or not, or +/- IDO1 inhibitor would be useful to look at the impact on T cell kynurenine uptake.

We understand that the reviewer surmised that a key goal and benefit of the fluorescence assay is the ability to analyze kynurenine in mixed cell populations and even endogenous kynurenine *in vivo*.

- We apologize for the lack of clarity here as we clearly failed to emphasize adequately that this kynurenine uptake assay measures the kynurenine transport POTENTIAL of cells – giving a readout of the cells capacity to transport a substrate. We make use of kynurenine as the transport substrate because it is a) a validated System L substrate and b) has spectral properties compatible with flow cytometric analysis, allowing this assay to be used on a single cell platform. – This is a FIRST for amino acid uptake assays, which previously have been constrained to population based analyses (radiolabeled uptake, or even metabolite analysis).

However, we do not wish to imply that all of the observed fluorescence in the V450 channel is due to kynurenine. Indeed, that is why we have included the cold competition controls (leucine or lysine) and the transporter inhibitor controls (BCH). Without these controls to define the increased fluorescence as coming from a transported substrate, all we are looking at is fluorescence. Thus, using this assay and assuming that any cell that has fluorescence in the V450 channel has taken up kynurenine “*in vivo*” would not be appropriate : it would be impossible to tell if fluorescence were due to transported kynurenine, or any other potentially auto-fluorescent metabolite or protein, without “transporter” controls.

Hence, the high levels of Kynurenine referred to are ONLY when used as a substrate for the transport assay.

The reviewer suggests looking at *in vivo* kynurenine in T cells in an IDO expressing tumour model. However, this is not possible with this assay/technique (for the reasons expressed above).

We hope this clarifies the issue for the reviewer.

Given the role of AhR in Th17 and Treg balance, it would be interesting to also look at Treg and show how kynurenine uptake may be regulated in this subset.

- The reviewer raises an interesting point. We feel that exploring the role of kynurenine on regulatory T cells is beyond the scope of the present work. However, from our proteomics studies, we know that splenic regulatory T cells *ex vivo* do not express SLC7A5. Moreover, using the flow based kynurenine uptake assay we also know that *ex vivo* splenic CD4 cells do not transport kynurenine ie there is no kynurenine positive subpopulation that would correspond to any resident CD25 positive Tregs. Nevertheless, regulatory T cells are complex cells and are present in many different tissues; both induced Tregs versus 'natural' Treg populations. We do not have the information about amino acid transport in all these different Tregs and feel that it would not be appropriate to make generic comments about regulatory T cells and kynurenine at this point. A full study would be a body of work in its own right.
- We wish to emphasize that the main findings of this paper are to report that uptake of kynurenine is tightly controlled in T cells and regulated through expression of the System L transporter SLC7A5. We verify this through biochemical parameters by competition studies, inhibitor studies as well as using transporter knockout mice. Furthermore, we have described a novel assay which measures the System L transport potential of cells at a single cell level, which is compatible with standard flow cytometry methods. We therefore feel that further, more detailed studies as to the immunological importance of kynurenine signaling in the broader T cell repertoire would be the basis of a whole new study, and as such are outwith the scope of this current paper.

Also, Th17 cell uptake of kynurenine is measured, but does this correlate with IL17 production? Based on literature, AhR should decrease Th17 cells.

We understand the reviewer to be asking whether the uptake of kynurenine by TH17 cells correlates with IL17 production?

- We have previously published that T cells which do not express SLC7A5 do not differentiate into IL17 producing Th17 cells, nor do fully differentiated Th17 cells transport kynurenine (data presented herein). However, we cannot prove this is causative. Moreover, our data show that the ability of Th17 cells to transport kynurenine is dynamic: T cells polarized under Th17 conditions show maximal kynurenine uptake on day 3 of culture yet at this time point the frequency of cells that can make IL-17 is quite low. In contrast, T cells polarized under Th17 conditions for 5 days no longer transport kynurenine yet make high levels of IL-17. So we would say that there is no correlation.

Statistics throughout should be evaluated to provide indications of significance. Error bars are provided, but statistical tests and indications of significance are not.

- We apologize for omitting these statistics from the figures and now include them.

Referee #3

Reviewer #3 (Remarks to the Author): (in black)

This manuscript describes how activated T cells import the tryptophan catabolite kynurenine (Kyn) using a novel flow cytometric method to detect Kyn in individual T cells. The pretext is that Kyn was reported previously to regulate T cell responses by activating the aryl hydrocarbon receptor (AhR). This notion has been controversial because Kyn is a relatively weak AhR ligand. The major claims are that activated T cells import Kyn actively via the System L amino acid transporter SLC7A5 to incite AhR signaling. These new observations suggest that Kyn serves as a natural T cell suppressant by targeting activated T cells selectively, as previous work by the same group has shown that SLC7A5 is up-regulated only when T cells activate. Data presented supports these points and documents that the novel FACS-based assay described is a useful and robust method to detect Kyn uptake by activated T cells. This point notwithstanding, the study is strong on methodology but weak on biological significance, as key implications arising are not addressed in depth. Hence, appropriate attention to the points below has the potential to increase the scope and impact of this study.

Response to reviewer 3: (in red)

We would like to thank the reviewer for the many positive and insightful comments about our paper, and respond to points raised below.

Point 1:

Data shown in the right panel of Figure 2A suggest that all CTLs imported Kyn, Are System L (SLC7A5) transporters expressed uniformly by all cells in the 'CTL' population?

- Yes, we do believe these CTLs express uniformly high SLC7A5; the *in vitro* generated CTLs are a uniform population expanded in IL2. Indeed, flow cytometry staining of CD98, a subunit of the System L transporters, follows a tight, log normal distribution.

Also, is Kyn uptake selective for CTLs when mixed with naïve (resting) T cells before adding Kyn?

- This is a valid query as it pertains to the suitability of kynurenine uptake to be used as a single cell assay. Indeed, we have done this particular test as a control, and can confirm that CTLs take up much higher levels of kynurenine than naïve/resting T cells. The specificity of kynurenine uptake in mixed populations is also evident in the co-culture experiment described in Figure 4 where the non-T cell splenocytes, ie cells not responding to CD3/CD28 activation, do not take up kynurenine, and the only cells which do are the WT T cells (CD4 and CD8) and NOT the SLC7A5 null T cells (CD4 and CD8), this data is now included as a supplemental figure.

Point 2:

In Figure 2, the authors cite KM values for Kyn uptake in the uM range but do not comment if this is likely to be in the same range as physiologic levels of Kyn. The authors refer to this issue in the opening paragraph of their Discussion but do not address it. This is a key point, as it relates to the current controversy regarding the significance of AhR signaling mediated by Kyn in physiologic settings.

- We fully agree with the reviewer that this is a very important point. To try to address this we have included new data; a dose response of kynurenine activation driving Cyp1A1 mRNA expression in D3 TH17 cells (maximal transporter expression) (Fig 5G), and induction of Cyp1A1 mRNA in response to kynurenine (at 10uM) in the presence or absence of an AhR antagonist (Fig 5H). We hope this data will support our conclusion that kynurenine (*in vitro*) is an AhR ligand, and capable of driving AhR signaling at lower concentrations than previously used in the literature – with the caveat that the transporter is expressed on the cells responding to kynurenine. Importantly, we have expanded our discussion to address the physiological relevance of kynurenine as an AhR ligand.

Point 3:

In Figure 3, the authors show convincingly that all (CD8+) CTLs from mice infected with listeria (Lm) can import Kyn via System L. However, data presented raises additional questions regarding this model. First, did CTLs from Lm-infected mice show evidence of Kyn uptake in the absence of added Kyn? Shaded histograms shown in Fig 3B provide a hint that this may be the case.

- We apologise to the reviewer if we were not clear, the data in Figure 3 B do not refer to the kynurenine uptake assay, instead these data show CD44 surface expression on uninfected control CD8 T cells and D7 listeria infected CD8 cells to indicate the extent of CD8 activation in response to listeria infection.

However, in response to the reviewers query, we would be unable to address whether effector cells from Lm-infected mice show evidence of kynurenine uptake *in vivo* with this assay. The only way to address this is to isolate cells and use mass spectrometry approach analyze the cellular metabolite pool. We cannot assume that any basal fluorescence in the V450 channel would be due to the uptake of endogenous kynurenine. It could be other metabolites /endogenous proteins.

We apologize for the lack of clarity here as we clearly failed to emphasize adequately that this kynurenine uptake assay measures the transport *potential* of cells – giving a readout of the cells capacity to transport a substrate. We decided to make use of kynurenine as the transport substrate because it is a) a validated System L substrate and b) has spectral properties compatible with flow cytometric analysis, allowing this assay to be used on a single cell platform. – This is a FIRST for amino acid uptake assays, which previously have been constrained to population based analyses (radiolabeled uptake, or even metabolite analysis).

However, we do not wish to imply that any fluorescence in the V450 channel observed is due to kynurenine. Indeed, that is why we have included the cold competition controls (leucine or lysine) and the transporter inhibitor controls (BCH). Without these controls to define the increased the fluorescence as coming from a transported substrate, all we are looking at is fluorescence. Thus, using this assay directly “*in vivo*” would not be appropriate : it would be impossible to tell if fluorescence were due to transported kynurenine, or any other potentially auto-fluorescent metabolite or protein, without “transporter” controls.

Linked to this question, do CTLs import Kyn selectively *in vivo*, either due to IDO activity induced during Lm-infection or if Kyn is injected into Lm-infected mice?

- We re-iterate that we would not be able to measure the levels of *in vivo* kynurenine uptake with this assay (for the reasons explained above). However, this is a pertinent question, and will relate to the total amount of available kynurenine compared to alternative System L substrates. The detailed micro-environmental metabolomics techniques required to perform this kind of analysis is beyond the scope of this paper, however we have tried to address this point in the discussion.

Second, did CD4 T cells (Foxp3⁻) and Tregs (Foxp3⁺) from Lm-infected mice also import Kyn? These points are of considerable interest because previous reports showed that Lm-infections induce IDO activity and that AhR signaling acts on Tregs to promote their regulatory functions.

- We have not analysed kyn transport in Tregs responding to an immune challenge. This, also, is a very interesting aspect, however we feel that it is beyond the scope of this paper.

Point 4:

Data presented in Figures 4 & 5 show convincingly that SLC7A5 ablation in CD4 T cells abolishes Kyn uptake in CD4 T cells selectively and that imported Kyn triggers AhR signaling in CD4 T cells, respectively. However, no data is presented to address the biological significance of these findings, specifically regarding the reported ability of Kyn to suppress Th1 responses (e.g. IFN γ production) or to reinforce Treg-mediated suppression of effector T cells via AhR signaling. As the pretext for this study was to address how Kyn activates AhR signaling to suppress T cell responses despite being a weak AhR ligand, data addressing these points would enhance the biological significance of this study.

- We are very excited that the reviewer sees the scope and potential of this assay to be used as a tool to investigate these important biological questions ie. when and how kynurenine and/or AhR signaling impacts upon T cell responses? However, we wish to emphasize that the main findings of this paper are to report that uptake of kynurenine is tightly controlled in T cells and regulated through expression of the System L transporter SLC7A5. We verify this through biochemical parameters by competition studies, inhibitor studies as well as using transporter knockout mice. Furthermore, we have described a novel assay which measures the System L transport potential of cells at a single cell level, which is compatible with standard flow cytometry methods. We therefore feel that further, more detailed studies as to the immunological importance of kynurenine signaling in the broader T cell repertoire would be the basis of a whole new study, and as such are outwith the scope of this current paper.

Point 5:

In the Discussion, the authors speculate that Kyn may mediate suppression via AhR-independent mechanisms such as attenuating mTOR activation by limiting T cell access to essential amino acids. Is there any evidence to support these speculations?

- Indeed, it is possible *in vitro* to manipulate CTLs which have high mTORC1 activity dependent upon leucine bioavailability. Removing these cells from leucine results in a loss of mTORC1 activity, and subsequent refeeding CTL with leucine (100 μ M) re-establishes mTORC1 activity. If this is done in the presence of equal amounts of Kyn (100 μ M) this will impair restoration of mTORC1 activity. However, not only is this a highly manipulated scenario, it is also dependent upon unlikely high kynurenine concentrations.

We have performed a series of transporter competition assays to try to estimate the competitive pressure upon the SLC7A5 transporter and thus the likelihood of kynurenine transport acting to limit cellular bioavailability of specific amino acids in a physiological situation. These data indicate that the levels of kynurenine required to out-compete the much higher affinity substrates, including the essential amino acids LEU, TRP and MET, exceed physiological kynurenine levels measured to date (that we are aware of), and allow us to conclude that only in microenvironments of fierce nutrient competition would this particular scenario ever be plausible. In with current knowledge/published data available, we feel that this would be highly unlikely, please see below. We have now addressed this further in the discussion.

We hope our revisions to this paper have addressed the salient points raised by each reviewer, and once again thank the reviewers for their time and critical insight.

Reviewers' comments:

Reviewer #1 (Remarks to the Author):

The authors have addressed my concerns and I think their manuscript is now suitable for publication.

While I agree with Reviewer 2 that the novelty of this manuscript is mainly of a technical nature, I think the authors have a point emphasising the importance of their finding with respect to single cell analysis of system L amino acid transport.

I remain sceptical about the importance of this transport mechanism for kynurenine mediated AhR activation. As the interesting paper cited by the authors shows the activity of kynurenine is basically due to a trace metabolite with very high affinity. As this metabolite seems to form upon storage of kynurenine it is unlikely that it requires kynurenine transport into cells to be formed in vivo. Nevertheless, kynurenine has important AhR independent functions in immune tolerance for which the mechanism described in this paper are highly relevant.

Reviewer #2 (Remarks to the Author):

The authors have made arguments to circumvent my concerns and they are generally valid. It is disappointing that they did not perform some relatively simple experiments, such as measuring if Treg could take up kynurenine and if this were to depend on the same mechanism. Given the interest in kynurenine and IDO1 in cancer biology, that would have been an important addition.

Reviewer #3 (Remarks to the Author):

The revised manuscript and rebuttal by Sinclair et al addresses some, but not all of my original concerns, as detailed below.

Point 1:

A new dataset (Supplemental Figure 1) was provided in response to my request for further clarification about the authors' claim to detect selective uptake of Kyn by CTLs (activated T cells) but not resting T cells. In rebuttal, the authors, "confirm that CTLs take up much higher levels of kynurenine than naïve/resting T cells". However, the new dataset does not address this issue, as it shows that all activated T cells take up Kyn (with CD8 T cells taking up more Kyn than CD4 T cells) but does not compare resting with activated T cells. Data reported in Figure 4 also fails to address this point, as data reported are from activated T cells from WT and Slc7A5-deficient mice; in other words, no data addresses if Kyn is taken up selectively by activated but not resting WT T cells in mixed cultures.

Point 3:

I accept the authors' clarifications that they did not (and cannot) test if Kyn was taken up by CTLs in the Listeria model. I also accept the authors' emphasis on the novelty of being able to measure transport potential of T cells. However, the relevant section (lines 154-194) of the manuscript has not been revised and remains misleading on this point, as measuring Kyn transport capacity of T cells in vivo is mentioned in several places. I had also assumed (incorrectly it now appears) that the authors selected the listeria model because several studies show that listeria infection induces IDO activity and Kyn production in vivo. The authors also stated that measuring Kyn transport in Tregs is beyond the scope of the present study (the same rebuttal point is also directed at a related question about Tregs posed by Reviewer 2). For these reasons, the authors' rebuttal on this point does not reduce but increases my original concern that the manuscript is quite weak on biological significance (see Point 4 below).

Point 4:

The authors provide no additional data to address the concern that the manuscript is weak on biological significance expressed by me and Reviewer 2.

Reviewers' comments:

Reviewer #1 (Remarks to the Author):

The authors have addressed my concerns and I think their manuscript is now suitable for publication.

While I agree with Reviewer 2 that the novelty of this manuscript is mainly of a technical nature, I think the authors have a point emphasising the importance of their finding with respect to single cell analysis of system L amino acid transport.

I remain sceptical about the importance of this transport mechanism for kynurenine mediated AhR activation. As the interesting paper cited by the authors shows the activity of kynurenine is basically due to a trace metabolite with very high affinity. As this metabolite seems to form upon storage of kynurenine it is unlikely that it requires kynurenine transport into cells to be formed *in vivo*. Nevertheless, kynurenine has important AhR independent functions in immune tolerance for which the mechanism described in this paper are highly relevant.

We accept the opinion/scepticism of reviewer 2 that kynurenine (as is) is the direct driver of AHR signalling. However, in our hands, whether it be through “pro-ligand” delivery or kynurenine itself, it is clear that SLC7A5 mediated transport is required in order for kynurenine to activate the AHR in T cells. We have demonstrated this (figure 5) by showing that only T cells that have high levels of System L transport are able to induce AHR signalling, and by competing kynurenine transport with a System L competitive blocker we impede AHR signalling. Whether the AHR activation is via kynurenine itself, or by the more highly active trace metabolite, the mechanism still holds true. We have expanded this in the discussion.

Furthermore, we have also emphasised in the discussion that transport of kynurenine will undoubtedly regulate any non-AHR dependent immune-modulatory functions. We do think our paper highlight a very important fact that for KN to do anything it needs to be transported.

Reviewer #2 (Remarks to the Author):

The authors have made arguments to circumvent my concerns and they are generally valid. It is disappointing that they did not perform some relatively simple experiments, such as measuring if Treg could take up kynurenine and if this were to depend on the same mechanism. Given the interest in kynurenine and IDO1 in cancer biology, that would have been an important addition.

The reviewer initially remarked that “given the context of the role of AhR in Th17 and Treg balance it would be interesting to also look at Treg and show how kynurenine uptake may be regulated in this subset”.

We did seriously consider these comments. But we have used high resolution mass spectrometry to map the proteomes of *ex vivo* Tregs and induced (*in vitro* expanded) Tregs. Neither express detectable AHR. Nor do *ex vivo* Tregs or induced (*in vitro* expanded) Tregs express high levels of SLC7A5 or even SLC7A6 (a “sister” transporter with many similar substrate specificities). Furthermore, we have checked, using Foxp3-GFP expressing mice and the uptake assay described in the paper, whether *ex vivo* Tregs take up kynurenine. We

find that these *ex vivo* Tregs do not take up kynurenine above other naïve CD4 cells. We are happy to share this data with the reviewer if requested.

The data in the literature regarding AhR and Tregs is complex and controversial: with some papers deriving no direct effects of AhR ligand activation and most papers describing data that DO define effects of AhR activation on/in Tregs looking at long-term cultures (or even whole body systems) where it is very difficult to be certain or identify which cells are the AhR ligand target. With this in mind, we did not wish to get embroiled in what is already a very complicated and complex field, feeling that this is beyond the scope of this current study.

Reviewer #3 (Remarks to the Author):

The revised manuscript and rebuttal by Sinclair et al addresses some, but not all of my original concerns, as detailed below.

Point 1:

A new dataset (Supplemental Figure 1) was provided in response to my request for further clarification about the authors' claim to detect selective uptake of Kyn by CTLs (activated T cells) but not resting T cells. In rebuttal, the authors, "confirm that CTLs take up much higher levels of kynurenine than naïve/resting T cells". However, the new dataset does not address this issue, as it shows that all activated T cells take up Kyn (with CD8 T cells taking up more Kyn than CD4 T cells) but does not compare resting with activated T cells. Data reported in Figure 4 also fails to address this point, as data reported are from activated T cells from WT and Slc7A5-deficient mice; in other words, no data addresses if Kyn is taken up selectively by activated but not resting WT T cells in mixed cultures.

Whilst the data in Supplemental fig 1 of CD4 and CD8 differences may NOT address naïve versus activated cell uptake – it does highlight that it is possible to measure quantitative differences of System L dependent uptake in different populations of cells within the same sample. – previously this has NOT been possible.

To address the specific question as to whether kynurenine uptake is greater in active T cells compared to naïve, in the same sample, I now include the simple "mix" of naïve and *in vitro* differentiated effector T cells (as asked). A caveat not to be overlooked with this data is that the background/intrinsic autofluorescence of these cells is vastly different. This is good to show as it highlights the importance of internal controls to truly assess uptake and NOT autofluorescence when using this assay.

Moreover, I also include a more "real life" experimental scenario; following kynurenine uptake *ex vivo* in adoptively transferred OT2 T cells after NP-OVA/alum immunisation compared with host (non-Ova specific) CD4 T cells.

I hope that this data together will adequately address the point that you can use this assay to determine differences in System L uptake capacity in different cells within the same population.

Point 3:

I accept the authors' clarifications that they did not (and cannot) test if Kyn was taken up by

CTLs in the Listeria model. I also accept the authors' emphasis on the novelty of being able to measure transport potential of T cells. However, the relevant section (lines 154-194) of the manuscript has not been revised and remains misleading on this point, as measuring Kyn transport capacity of T cells *in vivo* is mentioned in several places.

We thank the reviewer for this useful perspective and have modified the paper to clarify this point, that we are measuring the transport capacity *ex vivo* of *in vivo* activated T cells.

I had also assumed (incorrectly it now appears) that the authors selected the listeria model because several studies show that listeria infection induces IDO activity and Kyn production *in vivo*. The authors also stated that measuring Kyn transport in Tregs is beyond the scope of the present study (the same rebuttal point is also directed at a related question about Tregs posed by Reviewer 2). For these reasons, the authors' rebuttal on this point does not reduce but increases my original concern that the manuscript is quite weak on biological significance (see Point 4 below).

Point 4:

The authors provide no additional data to address the concern that the manuscript is weak on biological significance expressed by me and Reviewer 2.

We understand why the reviewer might feel this but our paper was not meant to be an exploration of kynurenine biology but rather a rigorous focused analysis of a fundamental issue of how T cells sense and transport this metabolite. We feel that our paper achieves 2 important advances that we wish to disseminate to the wider community:

- We highlight an underappreciated level of regulation for the tryptophan metabolite kynurenine with regard to its immunomodulatory action.
- We introduce a single cell assay to monitor amino acid uptake through System L amino acid transporters – this is an important addition to the investigative toolbox and therefore relevant to a broad biological community.